# Molecular Survey of Metastrongyloid Lungworms in Domestic Cats (*Felis silvestris catus*) from Romania: A Retrospective Study (2008–2011)

**DOI:** 10.3390/pathogens9020080

**Published:** 2020-01-26

**Authors:** Adriana Gyӧrke, Mirabela Oana Dumitrache, Zsuzsa Kalmár, Anamaria Ioana Paştiu, Viorica Mircean

**Affiliations:** 1Department of Parasitology and Parasitic Diseases, University of Agricultural Sciences and Veterinary Medicine Cluj-Napoca, 3-5 Calea Mănăştur Street, 400372 Cluj-Napoca, Romania; mirabela.dumitrache@usamvcluj.ro (M.O.D.); zsuzsa.kalmar@usamvcluj.ro (Z.K.); oana.pastiu@usamvcluj.ro (A.I.P.); viorica.mircean@usamvcluj.ro (V.M.); 2Department of Genetics and Hereditary Diseases, University of Agricultural Sciences and Veterinary Medicine Cluj-Napoca, Faculty of Veterinary Medicine, 3-5 Claea Mănăștur Street, 400372 Cluj-Napoca, Romania

**Keywords:** cats, respiratory nematodes, PCR, Romania

## Abstract

Background: Lungworms are recognized as important agents in the pathology of the respiratory system in domestic cats. While *Aelurostrongylus abstrusus* is worldwide known and studied, *Troglostrongylus brevior* has gained the attention of the scientific community only in the last decade. The pathogenicity of this species seems to be higher than *A. abstrusus*, causing more severe clinical presentations and being potentially fatal, especially in young animals. Methods: In this study, 371 DNA isolates of faecal samples were tested by multiplex polymerase chain reaction for the presence of *A. abstrusus*, *T. brevior,* and *Angiostrongylus chabaudi*. Results: The results showed that 30.2% and 6.7% of the investigated domestic cats were positive for *A. abstrusus* and *T. brevior* respectively, stressing out the importance of these parasites as agents of respiratory conditions in domestic cats from Romania. None of the samples were positive for *A. chabaudi*. The age, the outdoor access, and the lack of deworming were identified as significant risk factors for infection with *A. abstrusus.* Conclusions: This paper represents the first report of *T. brevior* in domestic cats from Romania. Moreover, it presents the most northern localization in Europe of *T. brevior*.

## 1. Introduction

In veterinary practice, lungworm infections in cats are of particular importance for the differential diagnosis of various respiratory diseases, being an important etiological agent of these conditions. The nematodes affecting the respiratory system of felids are represented by metastrongyloid worms *Aelurostrongylus abstrusus* (Strongylida, Angiostrongylidae), *Troglostrongylus* spp. (Strongylida, Crenosomatidae), *Angiostrongylus chabaudi* (Strongylida, Angiostrongylidae), *Oslerus rostratus* (Strongylida, Filaroididae), and the trichurid *Capillaria aerophila* (syn. *Eucoleus aerophilus*) (Trichurida, Capillariidae) [1]. *A. abstrusus* and *C. aerophila* are the most well-known, studied, and prevalent lungworm of domestic cats (*Felis silvestris catus*), causing infections in cats worldwide [2]. However, since the first report, in 2010 when it was identified in two cats out of seven in Spain [3], *T. brevior* triggered the attention of scientists and it became more sought. Moreover, this parasite was frequently diagnosed in kittens with respiratory signs in Italy and it was found that it can be fatal [4,5,6]. *A. chabaudi* is a cardiopulmonary nematode, localized to the right side of the heart and pulmonary arteries of the wild cats [7,8]. Even the parasite was isolated from two dead domestic cats in Italy [9,10], no natural patent infection in this host was described, stressing out if *A. chabaudi* can complete its biological cycle in domestic cats [11].

The routine diagnosis of metastrongyloid lungworm infection is based mainly on coproscopy, using the Baermann technique, followed by the morphological identification of L1 larvae. However, the differentiation between *A. abstrusus* and *T. brevior* is difficult due to overlapping morphological features and individual variations [2,12]. The L1 stage of these two species can be differentiated based on morphological characteristics of L1’s anterior and posterior end [12], but variations in the shape of the tail were detected within the same species [2]. Currently, L1 larvae of the metastrongyloid lungworms can be also specifically identified by molecular techniques [13,14].

Recently, *T. brevior* and *A. chabaudi* were identified in wild cats from Romania [15,16,17]. In the light of these new reports in our country and considering the importance of these agents in the respiratory pathology of cats, we aimed to perform a retrospective study to evaluate the prevalence of metastrongyloid lungworms (*A. abstrusus, T. brevior,* and *A. chabaudi*) in domestic cats from Romania using molecular techniques.

## 2. Results

### 2.1. Animals

The age of the animals ranged between one month to 17 years old, with an average age of 2.7 ± 3.0 years (32.8 ± 35.77 months). The ratio between females (*n* = 184) and males (*n* = 187) was 0.98. The animals originated from rural (*n* = 115) and urban (*n* = 260) areas. Of the total of 260 urban cats, 89 were indoor cats. 

Prophylactic anthelminthic treatments were performed in 30.5% of the investigated cats, with different frequencies (one, two, three, or four times per year; *p* = 0.4691) using various commercial products (Cestal Cat^®^, Drontal Cat^®^, Pratel^®^, Biheldon^®^, Caniverm^®^, and Stronghold^®^). The percentage of prophylactic anthelminthic treatments was significantly lower in cats originating from rural areas (13.1%) in comparison with the ones from urban areas (37.7%) (*p* < 0.0001). 

### 2.2. Descriptive Epidemiology

Overall, 31.5% (117/371; 95% CI: 26.9–36.6) of the examined cats were infected with at least one species of metastrongyloid lungworm. There were identified infections with *A. abstrusus* and *T. brevior*. 

One-hundred-twelve (30.2%; 95% CI: 25.6–35.2) out of 371 tested cats were positive for *A. abstrusus* by mPCR. The geographical origin of positive individuals is available in Figure 1. Animals with outdoor access (36.2%; *p* < 0.0001), from rural area (41.2%; *p* = 0.002), or without anthelmintic treatment (39.9%; *p* < 0.0001) presented statistically significant higher prevalence compared with indoor (11.2%), urban (25.3%), and dewormed (8.0%) cats, respectively (Table 1). The frequency of anthelmintic treatments did not influence (*p* = 0.767) the prevalence of *A. abstrusus* in prophylactically treated cats. No statistically significant differences were registered regarding *A. abstrusus* infections in cats, according to age and gender by bivariable analysis (Table 1). 

*T. brevior* DNA was identified in 25 (6.7%; 95% CI: 4.5–9.9) out of 371 samples by mPCR, from which six were confirmed by sequencing. All positive cats had outdoor access and most of them originated from urban areas and were not treated with antihelmintic products (Table 1). Most of the positive samples confirmed by sequencing (4/6) were registered in kittens (4.2%; 95% CI: 1.6–10.2). *T. brevior* was identified by mPCR in four countries (Harghita, Cluj, Hunedoara, and Dolj) and by sequencing it was confirmed in three out of 16 studied countries (Harghita, Cluj, and Hunedoara) (Figure 1). The co-infections with *A. abstrusus* were registered in 20 out of 25 positive samples (by mPCR) and in two out of six in the case of positive samples confirmed by sequencing (data not showed).

### 2.3. Risk Factors

According to multilogistic regression analysis, the age (*p* = 0.005), the outdoor access (*p* = 0.0004), and the lack of deworming (*p* = 0.0000) were identified as significant risk factors for infection with *A. abstrusus* in domestic cats from Romania (Table 1). The odds for cats older than six months and untreated ones were 1.52 (95% CI: 1.13–2.03) and 0.12 (0.06–0.27) times, respectively higher than in the case of kittens and dewormed cats. Cats with outdoor access were 4.19 (95% CI: 1.90–9.21) times more likely to be infected with *A. abstrusus* than indoor cats. 

Multivariable analyses of *T. brevior* positive samples showed that there are no statistically significant differences (Table 1).

### 2.4. Sequencing and Phylogenetic Analysis

The sequencing analysis of mPCR positive *T. brevior* samples revealed that six out of 25 samples presented an identity with this species. Four specimens (acc. no. MG978109–MG978110) out of six had 100% similar degree identity with *T. brevior*, while two specimens (acc. no. MG978111, MG978112) presented 99% identity. For the remaining 19 sequenced samples, the results showed a mixed template pattern and the BLAST analysis showed different degrees of identity with *A. abstrusus* (15 samples 80%–99%) or no result was obtained (*n* = 4). The phylogenetic analysis of the sequences clustered the specimens near *T. brevior* isolated from *Felis silvestris* (Luxembourg)*, F. silvestris catus* (Italy) (MG978111-MG978112), respectively from *Prionailurus bengalensis euptilurus* (Japan) and a cat in Italy (MG978110). One sequence, MG978109, clustered alone before all isolates available in GenBank (Figure 2). 

## 3. Discussion

Lungworms are recognized as important etiological agents in the pathology of the respiratory system in domestic cats and considered as emerging parasites in Europe [18,19]. Infection with lungworms in kittens can be fatal [4,5,20,21,22] while in elder cats, most cases are asymptomatic. However, infected animals are of great importance in maintaining the parasite in a certain territory.

Among respiratory nematodes, *A. abstrusus* is the most prevalent and widespread species parasitizing in cats, being highly recognized and studied. Regarding *T. brevior* and *A. chabaudi,* the interest is more recent and epidemiological data are scant, especially in certain countries. The first description of *T. brevior* was made in 1949 by Gerichter [23], in two wild felids, *Felis silvestris lybica*, (formerly Felis ocreata), and *Felis chaus*, (formerly Catolynx chaus), caught in the vicinity of the Dead Sea. The parasite has been rediscovered starting with 2010, being reported in domestic and wild cats, from different Mediterranean countries (Spain, Italy, Bosnia and Herzegovina, Bulgaria, Greece, Cyprus) [2,3,4,24,25,26,27]. *A. chabaudi* is more frequently reported in wild [7,8,11,15,17] than in domestic cats [9,10].

*Felis silvestris silvestris* (European wild cats) is considered the reservoir host of *T. brevior* and a major factor for the dissemination of the parasite [28].

In Romania, only a few epidemiological studies regarding pulmonary nematodes of cats are available, presenting data obtained by the usage of classical coproparasitologic methods. Past data revealed and confirmed the infection with *A. abstrusus* in domestic cats from Romania, but failed to demonstrate the presence of *T. brevior* or *A. chabaudi.*

The studies conducted in our country so far showed prevalence values for *A. abstrusus* ranging between 3.7%–14.2% [2,29,30]. The results of the present survey revealed a much higher prevalence (30.2%) of the same parasite. However, some of the methods used in these previous studies, coproparasitologic exams such as flotation (5.6%) [29] have diagnostic limits, which could explain the differences in results. Moreover, the sampled population in the latest two studies was represented mainly by urban cats that are usually more taking care of, including the prevention perspective (anthelmintic products). Nevertheless, samples used in the current study were collected between 2008 and 2011, in a period when in Romania the pet owners’ veterinary care was limited. In the last decade, this aspect has been improved and owners have become more aware and rigorous in which concerns the necessity to prevent the various medical condition (authors’ observation).

In Europe, the prevalence of *A. abstrusus* ranges between 0.8% and 50.0% [2,27,30,31], varying by latitude (it decreases from the south to the north), intermediate and paratenic hosts availability in a certain geographic region, sampled population (owned versus stray cats; indoor versus outdoor cats), and diagnostic method (flotation, Baermann, PCR, ELISA). The highest prevalence rates were reported in South-eastern and Central European countries as Albania (39.7% domestic cats; 50.0% free-roaming cats) [31,32], Bulgaria (27.5% domestic cats), and Hungary (22.5% domestic cats) [2]. All these data were obtained by the Baermann method or necropsy. In most Northern European countries *A. abstrusus* infection is sporadically reported. For instance, in Sweden, the first report of *A. abstrusus* in outdoor cats was recently published [33].

In the present study age, outdoor access and lack of deworming were identified as risk factors for the infection with *A. abstrusus.* Elder cats seem to be more susceptible to the infection with this nematode. This aspect has been previously noticed also in other studies [2,27,34]. The outdoor access was another factor found to facilitate the infection with *A. abstrusus*. Similarly, animals with rural origin (41.2%) presented a significantly higher prevalence of infection than urban cats (25.3%). Rural animals will likely have a better opportunity of getting in contact with intermediated or paratenic hosts. Other studies have identified the same trends [29,30,35,36]. Even, lack of deworming was identified as a risk factor, the frequency of prophylactic anthelminthic treatments (1–4 times/year) did not influence the prevalence of *A. abstrusus* (*p* = 0.767). Most of the dewormed cats were prophylactically treated with a combination of pyrantel + praziquantel products. It is well known that an efficient treatment of lungworms infection involves repeated check-ups and repeated treatments if necessary. Moreover, the recommended drugs for the treatment of feline aelurostrongylosis are fenbendazole in high doses, moxidectin, emodepside, selamectin, eprinomectin, and milbemycin oxime [1,37]. Veterinarians should perform coproparasitological exams using the Baerman technique in any cat with respiratory signs, even in indoor cats or in dewormed ones.

After the first report in 1949 [23] when *T. brevior* was found in an African wildcat and a jungle cat from Palestine, until 2010 there is only one other record of this parasite, in a wild cat from Italy [38]. Subsequently, Jefferies et al. (2010) [3] described the nematode parasitizing in domestic cats from Ibiza Island, Spain, drawing attention to *T. brevior* and stimulating the interest of the scientific community on this topic. Since then, it has been increasingly described in domestic cats from European Islands [4,24,34] and continental areas [20,35,39]. After consideration for a long time, an occasional parasite affecting only wild felids [40], *T. brevior* is now recognized as causative agents of respiratory disease in domestic cats. In kittens and cats less than one year of age, *T. brevior* can cause more severe clinical presentations than aelurostrongylosis, and can be potentially fatal [6,27]. 

Recently, *T. brevior* was identified in a wild cat from Romania (Crasna, Covasna county), becoming the most northern location of the parasite in Europe [16]. However, no other data regarding the presence of this parasite in domestic cats from Romania are available. The current study revealed the presence of *T. brevior* by analyzing 371 individual samples collected from cats originating from 16 counties in Romania, between 2008 and 2011. The total prevalence of 6.7% was found by using mPCR, with six (1.6%) of the samples being confirmed by sequencing. To the best of our knowledge, this is the first record of *T. brevior* in domestic cats from Romania and demonstrates that *T. brevior* was present in our territory before it was described in 2017 in the wild cat [16]. The results are comparable with the one obtained in Sardinia (Italy) (6.5%) by Tamponi et al. [34] or by Cavalera et al. [27] in Bari, Messina, and Siena (Italy) (8.7). Although the multivariable statistical analyses failed to identify the risk factors in the case of *T. brevior* infection, from the fact that all positive animals had outdoor access, this allows us to conclude that free-ranging cats are more susceptible to getting this nematode. The majority of positive cats confirmed by sequencing (4/6) were less than six months of age, underlying the idea that troglostrongylidosis is a pediatric respiratory disease [6,27]. Moreover, this observation is important from the clinical point of view, stressing out the importance of using specific pharmaceutical compounds in treating/preventing pulmonary nematodes in kittens and reducing the contamination of the environment with larval stages. 

This molecular study, as the study performed by Di Cesare et al. in Italy [35], failed to identify L1 larvae of *A. chabaudi* in faecal samples collected from domestic cats. 

## 4. Material and Methods

### 4.1. Animals and Samples

In this study, 371 DNA extracted from faecal samples, collected between 2008 and 2011, from domestic owned cats were included. The animals originated in 16 countries from Romania (Figure 1), living both in rural and urban environments. For each animal, information regarding age, breed, gender, address, outdoor access, and the number of anthelmintic treatments per year was registered. The faecal samples were individually collected. All animals that were living exclusively outdoor, were kept indoors for up to 48 h until the sample was obtained. If the cat did not defecate during the first 24 h, it was treated orally with lactulose in a dose of 5 ml/cat. All faecal samples were initially analyzed using ovoscopical methods and then the faeces were stored at −20 °C until further processing. The samples were originally collected with the aim of assessing the molecular prevalence of the apicomplexan parasite *Toxoplasma gondii*. As a consequence, no data regarding the presence of respiratory signs were registered. All samples were the subject of multiplex PCR analysis to identify the metastrongyloid lungworms *A. abstrusus, T. brevior,* and *A. chabaudi.*

### 4.2. DNA Extraction

Genomic DNA was extracted from 150 mg of each sample, using an Isolate faecal DNA kit (Bioline) following the manufacturer’s instructions.

### 4.3. Multiplex PCR

Multiplex polymerase chain reaction (mPCR) of internally transcribed spacer 2 (ITS2) region was used for simultaneous identification and discrimination of *A. abstrusus, T. brevior,* and *A. chabaudi*, following the protocol previously described [14].

DNA extracted from L1 *A. abstrusus* and *T. brevior* were used as the positive controls. L1 larvae of *A. abstrusus* were obtained from a positive cat, by the Baerman concentration method. L1 larvae of *T. brevior* were kindly supplied by Anastasia Diakou (Laboratory of Parasitology and Parasitic Diseases, School of Veterinary Medicine, Faculty of Health Sciences, Aristotle University of Thessaloniki, Greece). 

### 4.4. Sequencing and Phylogenetic Analysis

The *T. brevior* positive PCR products were purified using a FavorPrep GEL/PCR Purification mini kit (Favorgen Biotech Corp., Taiwan) and further sequenced (Macrogen Europe). Phylogenetic analysis was performed with the MEGA 7.0.26 software. The accession numbers of each lungworm species used in the phylogenetic tree construction are listed in Appendix A. The evolutionary history was inferred using the maximum-likelihood method based on the Tamura-Nei genetic model. The bootstrap consensus tree inferred from 1000 replicates is taken to represent the evolutionary history of the taxa analyzed.

## 5. Data Analysis

The study population was characterized according to (i) age (kittens: Less than six months; juveniles: Six to 12 months; adults: More than 12 months); (ii) gender (males and females); (iii) environment (urban and rural); (iv) outdoor access (no, yes); and (v) anthelmintic treatment (no, yes). 

Frequency, prevalence, and its 95% confidence interval were calculated for *A. abstrusus* and *T. brevior* infections. Risk factors expressed as odds ratio (OR) with 95% confidence interval were determined by bivariable analysis using the chi-square test and by multivariable logistic regression analysis. Established variables were introduced in the logistic regression models and they remained in the model if the likelihood ratio test of the model was significant (*p* ≤ 0.05). A *p*-value of ≤ 0.05 was considered statistically significant. All data analysis was performed with the EpiInfo 2000 software (Centers for Disease Control and Prevention: http://wwwn.cdc.gov/epiinfo/).

## 6. Conclusions

This is the first molecular survey of metastrongyloid lungworms in domestic cats from Romania and one of the few existing worldwide. To the best of our knowledge, this is the first report of *T. brevior* in domestic cats from Romania and the most northern report of this parasite in Europe. The analysis of the obtained data showed that age, outdoor access, and lack of deworming represent the main risk factors for infection with *A. abstrusus*. The frequency of prophylactic antihelminthic treatments did not influence the prevalence of feline lungworms. Taking into consideration the results of the current study, lungworms infection should be included in the differential diagnosis of respiratory diseases in cats, especially in those with outdoor access, regardless of the frequency of prophylactic antiparasitic treatments.

## Figures and Tables

**Figure 1 pathogens-09-00080-f001:**
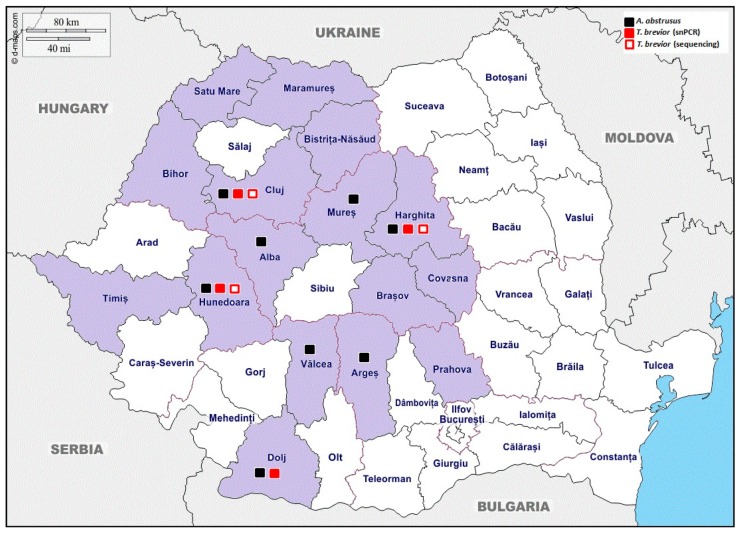
Geographical origin of sampled animals and positive cats for *A. abstrusus* and *T. brevior.*

**Figure 2 pathogens-09-00080-f002:**
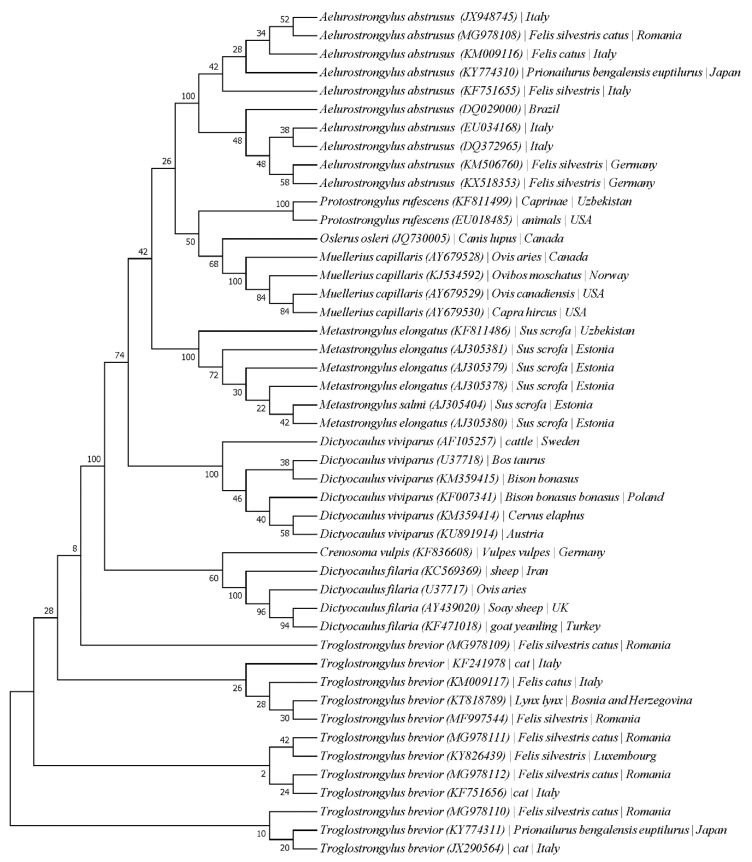
Phylogenetic analysis based on internally transcribed spacer 2 (ITS2) of *T. brevior*.

**Table 1 pathogens-09-00080-t001:** Frequency and prevalence of infection with *A. abstrusus* and *T. brevior* in domestic cats from Romania by age, gender, medium, habitat, and status of deworming.

	*A. abstrusus* (mPCR)			*T. brevior*					
				mPCR			Sequencing		
	Frequency	Prevalence (95% CI)	*p*	Frequency	Prevalence (95% CI)	*p*	Frequency	Prevalence (95% CI)	*p*
**Age (months)**									
0–6 (*n* = 96)	26	27.1 (18.5–37.1)	0.73	5	5.2 (1.7–11.7)	0.61	4	4.2 (1.6–10.2)	0.0595
6–12 (*n* = 65)	21	32.3 (21.2–45.1) **		6	9.2 (3.5–19.0)		1	1.5 (0.3–8.2)	
˃12 (*n* = 210)	65	31.0 (25.1–37.5) **		14	6.7 (3.7–10.9)		1	0.5 (0.0–2.7)	
**Gender**									
Females (*n* = 184)	56	30.4 (23.9–37.6)	0.99	14	7.6 (4.2–12.4)	0.65	2	1.1 (0.1–3.9)	0.70
Males (*n* = 187)	56	30.0 (23.5–37.1)		11	5.9 (3.0–10.3)		4	2.1 (0.6–5.4)	
**Medium**									
Rural (*n* = 114)	47	41.2 (32.6–50.4)	0.002	12	10.5 (5.6–17.7)	0.087	1	0.9 (0.2–4.8)	0.45
Urban (*n* = 257)	65	25.3 (20.4–30.9)		13	5.1 (2.7–8.5)		5	2.0 (0.8–4.5)	
**Habitat**									
Indoor (*n* = 89)	10	11.2 (5.5–19.7)	0.00001	0	0 (0.0–4.1)	0.008	0	0 (0.0–4.1)	0.37
Outdoor (*n* = 282)	102	36.2 (30.6–42.1) ***		25	8.9 (5.8–12.8)		6	2.1 (0.8–4.6)	
**Deworming**									
Yes (*n* = 113)	9	8.0 (3.7–14.6)	0.00000	3	2.7 (0.6–7.6)	0.064	1	0.9 (0.0–4.0)	0.77
No (*n* = 258)	103	39.9 (33.9–46.2) ***		22	8.5 (5.4–12.6)		5	1.9 (0.6–4.5)	
**Total (*n* = 371)**	112	30.2 (25.6–35.2)		25	6.7 (4.5–9.9)		6	1.6 (0.7–3.7)	

Logistic regression model: ** *p* < 0.01; *** *p* < 0.001.

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
