# Peer review of "Molecular Survey of Metastrongyloid Lungworms in Domestic Cats (Felis silvestris catus) from Romania: A Retrospective Study (2008–2011)"

_pathogens, 2020, doi:10.3390/pathogens9020080_

Round 1

Reviewer 1 Report

This is an interesting study building on previous work done across Europe to identify emerging respiratory parasites in domestic cats. The manuscript is well structured and the methodology clearly explained.

There are numerous grammatical errors scattered throughout the text which need to be addressed.

There is good reference to the published literature and the conclusions appear to be sound.

It isn't clear if ethical approval was required, sought or awarded for the work conducted.

Author Response

We would like to thank you for your comments. Bellow our answers.

(x) Moderate English changes required

Response: We have read one more time the manuscript and made the required English corrections.

This is an interesting study building on previous work done across Europe to identify emerging respiratory parasites in domestic cats. The manuscript is well structured and the methodology clearly explained.

Response: Thank you.

There are numerous grammatical errors scattered throughout the text which need to be addressed.

Response: We hope that we have done most of the grammatical corrections.

There is good reference to the published literature and the conclusions appear to be sound.

Response: Thank you. Anyway as reviewer 2 asked, we improved the conclusion section.

It isn't clear if ethical approval was required, sought or awarded for the work conducted.

Response: As we worked on the samples already collected between 2008 and 2011 an ethical approval wasn’t required.

Reviewer 2 Report

Interesting and needed topic of the article, is part of the current welfare trends of companion animals such as the domestic catThe abstract fully reflects the content of the full article. Keywords correctly chosen. The introduction without unnecessary descriptions informs about current methods of testing pulmonary parasites in cats and diagnostic difficulties associated with them. This explains the need for new molecular biology techniques to identify pulmonary parasite species. Description of results is clear and understandable. Unfortunately, the authors did not avoid mistakes.

- The description of the animals should include information on the castration of males and females. Interactions of sex hormones affect the overall immune status and hunting behavior, which in outdoor access cats can have a significant impact on lung parasite prevalence.

Lack of information whether the animals tested were asymptomatic carriers or demonstrated respiratory symptoms? If so, what and how were they diagnosed. Also  no information regarding faecal sampling. This is especially difficult for outdoor access cats and in rural areas. Unfortunately, there is no information about which antiparasitic preparation (active substance) were deworming and how often it was repeated. The method of deworming and the substances used affect the prevalence of parasitic invasions. In this situation, comparing dewormed and non-dewormed cats seems to have an unreliable result. In the obtained data showed that age and outdoor access (without  deworming) represent the main risk factors for the infection with A. abstrusus. In chapter "Conclusions", the sentence "The results highlight that pulmonary nematodes, A. abstrusus and T. brevior, could represent important agents of respiratory conditions in domestic cats from Romania." seems to be too vague, the more that the authors do not compare with other factors such as asthma for example. Also, the sentence from "Conclusions" chapter "If allowing the outdoor access represents the owner’s choice, veterinarians should promote the necessity of periodical deworming in cats with drugs chosen according to the age category and life style." is an over generalisation not related to the purpose of the work.  

Author Response

We would like to thank you for your comments and we appreciate them. We have made the changes suggested by you point by point and we believe our MS is now significantly improved.

Bellow our answers.

Are the methods adequately described? Can be improved

Response: We improved the Material and methods section as you suggested later.

Are the conclusions supported by the results? Can be improved

Response: We improved the Conclusion section as you suggested later.

Interesting and needed topic of the article, is part of the current welfare trends of companion animals such as the domestic cat. The abstract fully reflects the content of the full article. Keywords correctly chosen. The introduction without unnecessary descriptions informs about current methods of testing pulmonary parasites in cats and diagnostic difficulties associated with them. This explains the need for new molecular biology techniques to identify pulmonary parasite species. Description of results is clear and understandable.

Response: Thank you.

Unfortunately, the authors did not avoid mistakes.

- The description of the animals should include information on the castration of males and females. Interactions of sex hormones affect the overall immune status and hunting behavior, which in outdoor access cats can have a significant impact on lung parasite prevalence.

Response: At the time of sampling we did not registered information about castration of males and females. We are quite sure that cats from rural area were not castrated; as regarding the urban cats maybe some of them were castrated, a small percentage. Therefore, as we did not initially collect these type of data we are not able to include such information. However, we do agree with your suggestion.

Lack of information whether the animals tested were asymptomatic carriers or demonstrated respiratory symptoms? If so, what and how were they diagnosed.

Response: As we analyzed samples that “…were originally collected with the aim of assessing the molecular prevalence of the apicomplexan parasite Toxoplasma gondii.” the presence of respiratory signs was not assessed. This information is now included in the “Materials and Methods” section.

Also no information regarding faecal sampling. This is especially difficult for outdoor access cats and in rural areas. 

Response: Cats living exclusively outdoor were kept indoor for up to 48 hours in order to obtain the faecal samples. If the cat did not defecate during the first 24 hours, it was treated orally with lactulose in dose of 5 ml/cat. This information is now added in the manuscript.

Unfortunately, there is no information about which antiparasitic preparation (active substance) were deworming and how often it was repeated. The method of deworming and the substances used affect the prevalence of parasitic invasions. In this situation, comparing dewormed and non-dewormed cats seems to have an unreliable result. 

Response: We added in the manuscript to the following sections:

2.1. Animals: “Prophylactic anthelminthic treatments were performed in 30.5% of the investigated cats, with different frequencies (one, two, three or four times per year; p=0.4691) using various commercial products (Cestal Cat®, Drontal Cat®, Pratel®, Biheldon®, Caniverm® and Stronghold®). The percentage of prophylactic anthelminthic treatments was significantly lower in cats originating from rural area (13.1%) in comparison with the ones from urban area (37.7%) (p<0.0001).”

2.2. Descriptive epidemiology: “The frequency of anthelmintic treatments did not influence (p=0.767) the prevalence of A. abstrusus in prophylactic treated cats.”

Discussion: “Even, lack of deworming was identified as a risk factor, the frequency of prophylactic anthelminthic treatments (1-4 times/year) did not influence the prevalence of A. abstrusus (p=0.767). Other studies have identified the same trends [29,30,35,36]. Most of the dewormed cats were prophylactically treated with a combination pyrantel+praziquantel products. It is well known that an efficient treatment of lungworms infection involves repeated check-ups and repeated treatments if necessary. Also, the recommended drugs for the treatment of feline aelurostrongylosis are fenbendazole in high doses, moxidectin, emodepside, selamectin, eprinomectin and milbemycin oxime [1,37]. Veterinarians should perform coproparasitological exam using Baerman technique in any cat with respiratory signs, even in indoor cats or in dewormed ones.”

In the obtained data showed that age and outdoor access (without  deworming) represent the main risk factors for the infection with A. abstrusus. In chapter "Conclusions", the sentence "The results highlight that pulmonary nematodes, A. abstrusus and T. brevior, could represent important agents of respiratory conditions in domestic cats from Romania." seems to be too vague, the more that the authors do not compare with other factors such as asthma for example. Also, the sentence from "Conclusions" chapter "If allowing the outdoor access represents the owner’s choice, veterinarians should promote the necessity of periodical deworming in cats with drugs chosen according to the age category and life style." is an over generalisation not related to the purpose of the work.  

Response: Thank you. The conclusions were improved as follows: “This is the first molecular survey of metastrongyloid lungworms in domestic cats from Romania and one of the few existing worldwide. From the best of our knowledge this is the first report of T. brevior in domestic cats from Romania and the most northern report of this parasite in Europe. The analysis of the obtained data showed that age, outdoor access and lack of deworming represent the main risk factors for infection with A. abstrusus. The frequency of prophylactic antihelminthic treatments did not influence the prevalence of feline lungworms. Taking in consideration the results of the current study, lungworms infection should be included in the differential diagnosis of respiratory diseases in cats, especially in those with outdoor access, regardless of the frequency of prophylactic antiparasitic treatments.”